# Macular Pigment Optical Density and Photoreceptor Outer Segment Length as Predisease Biomarkers for Age-Related Macular Degeneration

**DOI:** 10.3390/jcm9051347

**Published:** 2020-05-05

**Authors:** Norihiro Nagai, Sakiko Minami, Misa Suzuki, Hajime Shinoda, Toshihide Kurihara, Hideki Sonobe, Kazuhiro Watanabe, Atsuro Uchida, Norimitsu Ban, Kazuo Tsubota, Yoko Ozawa

**Affiliations:** 1Laboratory of Retinal Cell Biology, Keio University School of Medicine, 35 Shinanomachi, Shinjuku-ku, Tokyo 160-8582, Japan; nagai@a5.keio.jp (N.N.); misayutakatomo@icloud.com (M.S.); 2Department of Ophthalmology, Keio University School of Medicine, 35 Shinanomachi, Shinjuku-ku, Tokyo 160-8582, Japan; saki.love5@icloud.com (S.M.); shinoha@mac.com (H.S.); kurihara@z8.keio.jp (T.K.); betty_vol2@ybb.ne.jp (H.S.); gaku047nikoniko3mickey@yahoo.co.jp (K.W.); uchidats@gmail.com (A.U.); nban@keio.jp (N.B.); tsubota@z3.keio.jp (K.T.); 3Department of Ophthalmology, St. Luke’s International Hospital, 9-1 Akashi-cho, Chuo-ku, Tokyo 104-8560, Japan; 4St. Luke’s International University, 9-1 Akashi-cho, Chuo-ku, Tokyo 104-8560, Japan

**Keywords:** macular pigment, photoreceptor, age-related macular degeneration, retina, medical checkup, biomarker

## Abstract

To explore predisease biomarkers, which may help screen for the risk of age-related macular degeneration (AMD) at very early stages, macular pigment optical density (MPOD) and photoreceptor outer segment (PROS) length were analyzed. Thirty late AMD fellow eyes, which are at high risk and represent the predisease condition of AMD, were evaluated and compared with 30 age-matched control eyes without retinal diseases; there was no early AMD involvement in the AMD fellow eyes. MPOD was measured using MPS2^®^ (M.E. Technica Co. Ltd., Tokyo, Japan), and PROS length was measured based on optical coherence tomography images. MPOD levels and PROS length in the AMD fellow eyes were significantly lower and shorter, respectively, than in control eyes. MPOD and PROS length were positively correlated in control eyes (R = 0.386; *p* = 0.035) but not in AMD fellow eyes. Twenty (67%) AMD fellow eyes met the criteria of MPOD < 0.65 and/or PROS length < 35 μm, while only five (17%) control eyes did. After adjusting for age and sex, AMD fellow eyes more frequently satisfied the definition (*p* < 0.001; 95% confidence interval, 3.50–60.4; odds ratio, 14.6). The combination of MPOD and PROS length may be a useful biomarker for screening predisease AMD patients, although further studies are required in this regard.

## 1. Introduction

Recent progress in medical science has improved the prognosis of blinding diseases; however, age-related macular degeneration (AMD) remains a leading cause of blindness worldwide [1,2]. As the retina is a neural tissue and may sustain irreversible changes once damaged, preventive therapy is advised [3,4], and recommendations such as regular intake of micronutrient supplements and giving up smoking are emphasized in the guidelines for AMD prevention [5] (http://www.nichigan.or.jp/member/guideline/aging_macular_degeneration.pdf). These interventions are currently recommended for patients with early AMD, which is characterized by the presence of drusen and/or pigment abnormalities in the retina (i.e., fundus), and large-scale clinical studies, such as the Age-Related Eye Disease Study (AREDS) and AREDS2 [3,4], have shown that they were significantly effective in preventing progression to late AMD during 5 years of follow-up in patients with early AMD. However, even without progression to late AMD, early AMD lesions can already cause visual impairment [6,7]. Moreover, systemic management of metabolic syndrome, another risk factor for AMD [8], is best initiated as early as possible, given that AMD develops as a result of chronic pathological cycles of inflammation and oxidative stress [9,10,11]. Thus, if there are biomarkers that represent earlier pathological and predisease conditions than drusen and/or pigment abnormalities, at-risk individuals can initiate preventive actions at an earlier time point to further minimize the damage.

AMD is a bilateral disease, and the risk of developing late AMD in a fellow eye increases over time after initial diagnosis [12]. In the U.S. AREDS, 43% of fellow eyes developed late AMD within 5 years [3]. A study in the United Kingdom showed that the rate of fellow eye involvement was 32% 2 years after initial diagnosis and the median time interval to involvement was 71 weeks [11]. Therefore, AMD fellow eyes may have already acquired predisease characteristics, even if they do not exhibit drusen and/or a pigment abnormality. Thus, analyzing such measurable characteristics in AMD fellow eyes could help establish early biomarkers for AMD, which may be utilized for screening predisease and high-risk patients and for considering early intervention for AMD prevention.

Macular pigment optical density (MPOD) reflects the levels of macular pigment (MP), which is composed of lutein, zeaxanthin, and mezo-zeaxanthin. Lutein and zeaxanthin are included in the micronutrient supplements recommended by AREDS2 [4]. MP can protect the retina from daily light stress [4,13] by absorbing high-energy short-wavelength light [14] (450–500 nm, with peak absorption at 460 nm) [15] and acting as an antioxidant in the retina [16]; it reduces reactive oxygen species both by scavenging them and inducing antioxidative enzymes, and it also suppresses inflammatory mediators in the retina [17,18,19,20,21,22]. MPOD levels are low when dietary intake and serum levels of lutein are low; in such cases, the preventive effects of additional lutein administration via micronutrient supplementation on AMD progression have been clearly observed [4,17,23,24,25]. It is believed that lutein intake increases the serum and, subsequently, macular levels of lutein; therefore, its intake prevents AMD progression. In fact, MPOD levels are low in AMD patients [26] and, therefore, low MPOD could reflect one aspect of AMD risk.

AMD most likely affects visual function because it affects photoreceptor cells. Morphological photoreceptor changes are effectively analyzed by using optical coherence tomography (OCT). Apart from photoreceptor layer thickness and volume, disruptions of the external limiting membrane, ellipsoid zone, and interdigitation zone can also reflect AMD-related early photoreceptor changes [27,28]. In addition, the length of the photoreceptor outer segment (PROS), where the visual pigment is concentrated and which can be reduced by inflammation and oxidative stress [17,19,21,29,30,31,32], is also measurable in OCT images, although it has not been assessed in AMD. PROS length can be sensitive to pathological conditions involving photoreceptor cells; reduced PROS length is associated with worse visual outcomes in branch retinal vein occlusion [33], diabetic macular edema [34], central serous chorioretinopathy [35], and idiopathic epiretinal membrane [36].

In this study, we measured the MPOD and PROS length and compared the corresponding data between AMD fellow eyes and age-matched control eyes of subjects without retinal diseases. The AMD fellow eyes in the current study did not include early AMD eyes characterized by drusen and/or a pigment abnormality confirmed by medical records, fundus photographs, or OCT images. Nevertheless, such AMD fellow eyes are at a high risk of developing AMD [3], and may be defined as being in the “predisease stage” of AMD. Thus, the current results may reveal earlier biomarkers of AMD than drusen and/or pigment abnormalities as observed in fundus photographs and help establish sensitive indicators that can be used in a medical checkup.

## 2. Experimental Section

This study was conducted according to the guidelines laid down in the Declaration of Helsinki. All procedures involving human subjects were approved by the Ethics Committee of Keio University School of Medicine (Approval No. 20150011, registered as UMIN000017845). Written informed consent was obtained from all subjects.

### 2.1. Subjects

This study was conducted in the Medical Retina Division Clinic, Department of Ophthalmology, Keio University School of Medicine from March 2016 to April 2018 and included 30 unilateral AMD patients and 30 control patients who did not have retinal diseases or glaucoma. AMD patients with other retinal diseases or glaucoma were excluded. Presence or absence of AMD and other diseases were confirmed by medical records including ophthalmologic examinations and previous angiographies (for the AMD patients), fundus photographs, and OCT images acquired at the time of this study. Patients who had severe cataracts were excluded from the study.

### 2.2. Measurement of MPOD

MPOD was measured using a macular densitometer (Macular pigment screener MPS2^®^, M.E. Technica Co. Ltd., Tokyo, Japan) that employs a heterochromatic flicker photometry technique described elsewhere [37,38]. Briefly, the difference in the sensitivity to blue wavelength (465 nm; absorbed by macular pigment) and green wavelength (530 nm; not absorbed by macular pigment) flicker light that can be recognized in the fovea (where macular pigment is concentrated) was recorded to evaluate the level of macular pigment that filtered blue light and increases the threshold of responsiveness to blue light by the photoreceptor cells. Thus, if macular pigment levels were reduced, MPOD would be lower. MPOD measurement was performed with best correction, and by using a series of stimuli programmed by the manufacturer. Non-testing eyes were occluded. The values were adjusted by age according to the manufacturer’s empirical algorithm [39,40] and recorded as the estimated MPOD [37].

### 2.3. Analysis of PROS Length Using OCT

OCT was performed using a Heidelberg Spectralis OCT system (Heidelberg Engineering GmbH, Dossenheim, Germany) in the afternoon, from 13:00 to 17:00. The macular area was scanned, and it acquired a dense volume scan (20 × 20°, 6 × 6 mm); 97 B-scans, each spaced 60 μm apart; and an automatic real-time mean of 2 scans. The PROS length was defined as the distance between the inner border of the retinal pigment epithelium and the inner border of the ellipsoid zone at the fovea, as determined by the deepest retinal depression in the horizontal images of the three-dimensional dense volume OCT scan. Measurements were obtained using the built-in software of the OCT device.

### 2.4. Ophthalmologic Examinations

All included subjects underwent best-corrected visual acuity (BCVA) measurements using the refraction test, intraocular pressure measurement, and fundus examination and photographing.

### 2.5. Statistical Analysis

All results are expressed as the mean ± standard error. Commercial statistical software (SPSS; ver. 25, SPSS Chicago, IL, USA) was used for the analysis. Two-tailed *t*-tests, Pearson’s correlation coefficient analyses, chi-squared tests, and stepwise multiple logistic regression models were used. Differences were considered statistically significant at *p* < 0.05.

## 3. Results

### 3.1. Participants’ Characteristics

Thirty AMD fellow eyes of 30 late AMD patients (22 men; mean age, 68.2 ± 1.8 years; range 50–85 years) and 30 eyes of control patients who had no retinal diseases or glaucoma (16 men; mean age, 69.2 ± 1.5 years; range, 51–84 years) were included in the study (Table 1). There were no significant differences in age, sex, BCVA, refractive errors, or central retinal thickness between the two groups. All of the examined patients had mild nuclear and/or cortical cataracts, and no patient had highly myopic (<-6 diopters) or pseudophakic eyes.

### 3.2. Macular Pigment Optical Density (MPOD) and Photoreceptor Outer Segment (PROS) Length

The mean MPOD in AMD fellow eyes was significantly lower than that in control eyes (AMD fellow eyes, 0.66 ± 0.16; Control 0.75 ± 0.15; *p* = 0.025, Figure 1a). The mean PROS length in AMD fellow eyes was significantly shorter than that in control eyes (AMD fellow eyes, 35.2 ± 1.1 μm; Control 39.9 ± 0.9 μm, *p* = 0.002, Figure 1b).

Next, we analyzed the relationship between MPOD and PROS length in each group (Figure 2). In control eyes, there was a significant positive correlation between MPOD and PROS length (R = 0.386, *p* = 0.035), whereas there was no significant correlation between MPOD and PROS length in AMD fellow eyes (R = 0.011, *p* = 0.952).

### 3.3. Distribution of MPOD and PROS Length

Scatter analysis showed that MPOD < 0.65 was observed in 13 (43%) AMD fellow eyes and in 5 (17%) control eyes (data not shown, chi-square test, *p* = 0.024); PROS length < 35 μm was observed in 13 (43%) AMD fellow eyes and in 2 (7%) control eyes (data not shown, chi-square test, *p* = 0.001). Moreover, the combination of MPOD < 0.65 and/or PROS length < 35 μm was observed in 20 AMD fellow eyes (67%) and 5 control eyes (17%) (Figure 3, Table 2, *p* = 0.00009) with a much greater significant difference.

### 3.4. Combination of MPOD and PROS Length May Be Useful Biomarkers for Screening Predisease AMD

Finally, we analyzed whether MPOD and PROS length are related to a risk of AMD in fellow eyes using stepwise multiple logistic regression models. After adjusting for age and sex, eyes with MPOD < 0.65 (*p* = 0.023) and PROS length < 35 μm (*p* = 0.003) involved more AMD fellow eyes than control eyes. Moreover, when the two parameters were combined and the definition was set as MPOD < 0.65 and/or PROS length < 35 μm, involvement of AMD fellow eyes became clearer (odds ratio = 14.6; *p* = 0.0002; 95% confidence interval, 3.50 to 60.4; Table 3).

Figure 4 shows the representative optical coherence tomography (OCT) images of control and AMD fellow eyes with MPOD.

## 4. Discussion

In the present study, MPOD, which reflects macular pigments levels, and PROS length, which represents photoreceptor structure, showed significantly lower levels in the fellow eyes of AMD patients than in the eyes of control participants. MPOD in the eyes of subjects in the control group positively correlated with PROS length; however, there were no significant correlations between these parameters in the fellow eyes of AMD patients. With cutoff values of MPOD < 0.65 and PROS length < 35 μm, AMD fellow eyes met the definitions more frequently than control eyes. Moreover, the risk of involving AMD fellow eyes, and predisease eyes, was much more clearly shown by the combined definition of MPOD < 0.65 and/or PROS length < 35 μm.

The MPOD in AMD fellow eyes was significantly lower than the MPOD in control eyes, consistent with previous reports [26,37]. We have previously reported that pooled data of AMD fellow and AMD-affected eyes had lower MPOD values as measured by MPS2 compared with age-matched healthy eyes [37]. Similarly, Obana et al. showed that early AMD and late AMD patients had significantly lower MPOD values as measured using resonance Raman spectroscopy than those in normal subjects older than 60 years [26]. These results, as well as our current results, support the theory that low levels of MP may accelerate AMD progression and that increasing MP levels by micronutrient supplementation may slow progression [4].

Moreover, we showed that PROS length in AMD fellow eyes was significantly lower than that in control eyes. Analyses of OCT images of AMD patients have been previously reported; Lamin et al. showed that retinal nerve fiber layer and photoreceptor layer volumes decreased within two years [41], and Brandl et al. showed that the thickness of the retinal pigment epithelium/Bruch’s membrane complex increased and that of the photoreceptor layer decreased more in the eyes with early AMD than in those with no AMD [42]. However, PROS length has not been evaluated in eyes at high risk of AMD. The current study was the first study to demonstrate the change of PROS length in AMD fellow eyes that still do not have either early or late AMD but are at a high risk of progression to AMD in the future, and which therefore represent the predisease condition. The PROS is composed of several hundreds of infolded plasma membrane discs that contain visual pigments [43]. PROS change could be a sensitive biomarker of photoreceptor degeneration, which appears prior to photoreceptor death, although further study is required in this regard.

In the present study, there was a significant correlation between MPOD and PROS length in control eyes, but not in AMD fellow eyes. Histological analysis has shown that lutein is distributed not only in the inner plexiform layer, but also in the outer plexiform layer where photoreceptors form synapses with inner neurons [44,45]. PROS is a component of photoreceptors; thus, both MPOD and PROS length can theoretically reflect photoreceptor health. Both parameters were reduced in AMD fellow eyes; however, as can be observed in the scatter plot (Figure 3), some AMD fellow eyes already show evident abnormality only in terms of the MPOD, while others show it only in terms of PROS length. There may be variations in pathological courses in the predisease condition, and, therefore, MPOD and PROS length were not correlated in the predisease eyes. In the current study, we achieved clearer associations with the AMD predisease condition by combining both these potential biomarkers. This combination of the parameters might have overwhelmed the variation, and, thus, we achieved a clear definition. Moreover, the differences in the MPOD and PROS lengths with or without the AMD predisease condition were relatively small; thus, the combination of these parameters may have increased the accuracy. The combination of these parameters may be applicable in medical checkups to screen for potential risk of AMD, although further future studies are required.

One of the limitations of this study was the relatively small number of exclusively Asian participants. Furthermore, no early nor late AMD eyes were included. However, the purpose of this study was to explore predisease biomarkers; therefore, we included eyes with no significant retinal findings and otherwise healthy eyes of subjects with AMD in the other eye. Nonetheless, our results reveal potentially valuable biomarkers for screening for the predisease condition of AMD and distinguishing between at-risk and healthy individuals.

In summary, the combination of MPOD and PROS length may be a novel biomarker for screening for AMD risk, which could be applied in medical checkups and help in proposing or initiating preventive actions. Further large-scale studies are warranted to verify the findings of this study.

## Figures and Tables

**Figure 1 jcm-09-01347-f001:**
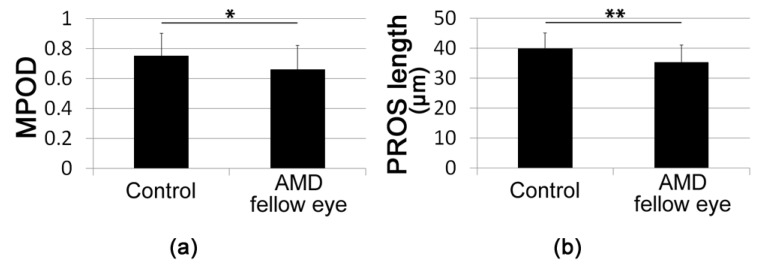
Macular pigment optical density (MPOD) and photoreceptor outer segment (PROS) length of control and age-related macular degeneration (AMD) fellow eyes. MPOD (**a**) was lower and PROS length (**b**) was shorter in AMD fellow eyes compared with age-matched control eyes. ** *p* < 0.01, * *p* < 0.05. Two-tailed *t*-test.

**Figure 2 jcm-09-01347-f002:**
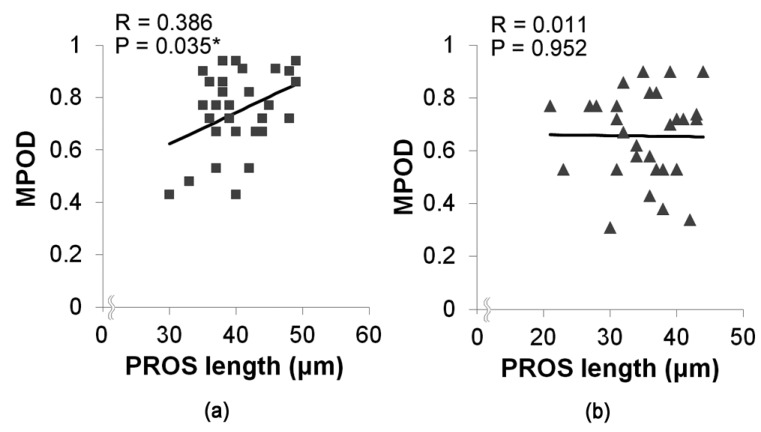
Correlation between macular pigment optical density (MPOD) and photoreceptor outer segment (PROS) length in control and age-related macular degeneration (AMD) fellow eyes. (**a**) In control eyes, there was a significant correlation between MPOD and PROS length (R = 0.386, *p* = 0.035). (**b**) In AMD fellow eyes, there was no significant correlation between MPOD and PROS length (R = 0.011, *p* = 0.952). Pearson’s correlation coefficient analyses; **p* < 0.05 ■Control eyes; ▲AMD fellow eyes.

**Figure 3 jcm-09-01347-f003:**
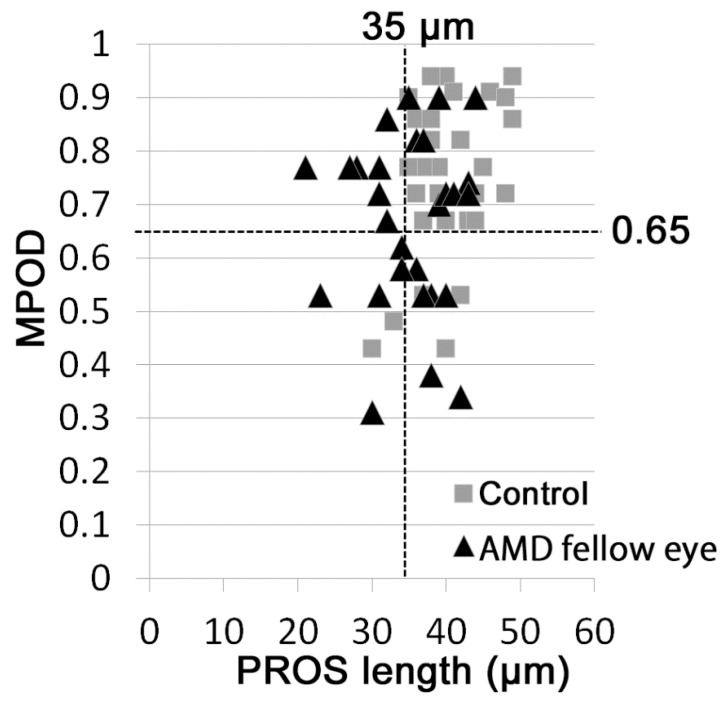
Scatter diagram of macular pigment optical density (MPOD) and photoreceptor outer segment (PROS) length in control and age-related macular degeneration (AMD) fellow eyes. MPOD and PROS length of control and AMD fellow eyes were plotted. ■Control eyes; ▲AMD fellow eyes.

**Figure 4 jcm-09-01347-f004:**
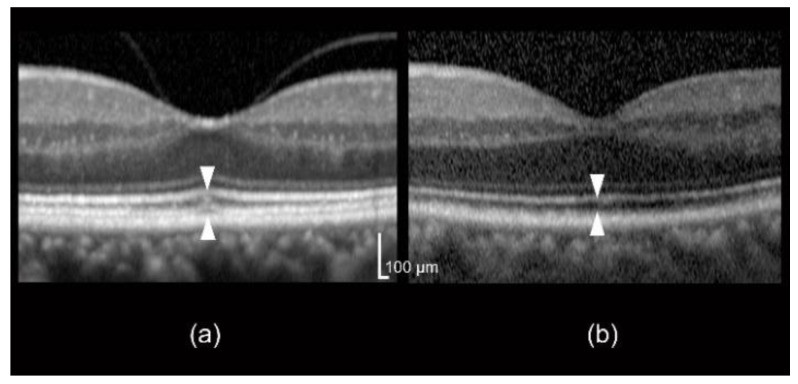
Representative optical coherence tomography (OCT) images of control and age-related macular degeneration (AMD)-fellow eyes. (**a**) Control eye; a 68-year-old man whose photoreceptor outer segment (PROS) length was 42 μm and macular pigment optical density (MPOD) was 0.82. (**b**) AMD fellow eye; A 54-year-old man whose PROS length was 37 μm and MPOD was 0.53. PROS length is shown by the distance between two arrowheads.

**Table 1 jcm-09-01347-t001:** Participants’ characteristics of control and age-related macular degeneration (AMD) fellow eyes.

	Control Eyes	AMD Fellow Eyes	*p*
n (eyes)	30	30	
Age (years)	51–84 (69.2 ± 1.5)	50–85 (68.2 ± 1.8)	0.678
Sex (men; eyes (%))	16 (53.3)	22 (73.3)	0.108
BCVA (LogMAR)	−0.08–0.05 (−0.06 ± 0.01)	−0.09–0.10 (−0.07 ± 0.01)	0.534
Refractive error (diopters)	−5.88 – +3.50 (−0.88 ± 0.49)	−4.63 – +3.50 (0.21 ± 0.33)	0.071
CRT (μm)	194−279 (226 ± 4)	120–270 (215 ± 5)	0.078
Data are presented as range (mean ± standard error); Two-tailed *t*-test. BCVA, best-corrected visual acuity; CRT, central retinal thickness.			

**Table 2 jcm-09-01347-t002:** Number of patients with macular pigment optical density (MPOD) < 0.65 and/or photoreceptor outer segment (PROS) length < 35 μm.

	Control Eyes	AMD Fellow Eyes	*p*
MPOD < 0.65 and/or PROS length < 35 μm	5 eyes (17%)	20 eyes (67%)	*p* < 0.001
MPOD ≥ 0.65 and PROS length ≥ 35 μm	25 eyes (83%)	10 eyes (33%)	
Chi-squared test. AMD, age-related macular degeneration; MPOD, macular pigment optical density; PROS, photoreceptor outer segment.			

**Table 3 jcm-09-01347-t003:** Involvement of age-related macular degeneration (AMD) fellow eyes according to macular pigment optical density (MPOD) and photoreceptor outer segment (PROS) length.

	OR	*p*	95% CI
MPOD < 0.65 and/or PROS length < 35 μm	14.6	0.0002	3.50 to 60.4
MPOD < 0.65	4.3	0.023	1.22 to 15.0
PROS length < 35 μm	13.6	0.003	2.36 to 78.3
Multiple logistic regression analysis adjusted for age and sex. MPOD, macular pigment optical density; PROS, photoreceptor outer segment.			

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
