# Peer review of "Macular Pigment Optical Density and Photoreceptor Outer Segment Length as Predisease Biomarkers for Age-Related Macular Degeneration"

_jcm, 2020, doi:10.3390/jcm9051347_

Round 1

Reviewer 1 Report

In this manuscript, the authors evaluated the MPOD and PROS between 30 fellow eyes from late AMD patients and 30 control eyes without retinal diseases. They demonstrated that MPOD/ PROS are lower /shorter in the fellow eyes from late AMD patients. They also showed that eyes with MPOD < 0.65 and PROS length < 35 µm involved more AMD-fellow eyes than control eyes. The authors concluded that the combination of MPOD and PROS length may be a biomarker for screening predisease AMD patients.

The study is well designed and the manuscript is well written. The results provided interesting information in the field. Some weaknesses are as below:

  1. MPOD and PROS changes are only about 12% compared to control group, although there is statistically significance.
  2. The sample size (30 eyes for each group) is relatively small.
  3. Have the authors measured and evaluated the MPOD and PROS between early-AMD fellow eyes and control eyes without retinal diseases?

Author Response

Point-by-Point Responses to the Reviewers’ Comments

Reviewer 1

Comments and Suggestions for Authors

In this manuscript, the authors evaluated the MPOD and PROS between 30 fellow eyes from late AMD patients and 30 control eyes without retinal diseases. They demonstrated that MPOD/ PROS are lower /shorter in the fellow eyes from late AMD patients. They also showed that eyes with MPOD < 0.65 and PROS length < 35 µm involved more AMD-fellow eyes than control eyes. The authors concluded that the combination of MPOD and PROS length may be a biomarker for screening predisease AMD patients.

The study is well designed and the manuscript is well written. The results provided interesting information in the field. Some weaknesses are as below:

We appreciate the reviewer for understanding our work.

  1. MPOD and PROS changes are only about 12% compared to control group, although there is statistically significance.

We agree that the MPOD and PROS changes were not substantial, although the differences were significant. This would be because we compared the values of control participants with predisease patients, and not with symptoms/signs-having patients. It would be a nature that prediseased, and not disease diagnosed, patients have only slight changes. To support this weak point, combination of 2 markers would be of value. We added this point in the discussion point as follows:

Line 228-

Moreover, the differences in the MPOD and PROS lengths with or without AMD-predisease condition was relatively small; thus, the combination of these parameters may have increased the accuracy.

  1. The sample size (30 eyes for each group) is relatively small.

We also agree this point, and we have already included this in the limitation paragraph of the original manuscript as follows;

Line 232-

One of the limitations of this study was the relatively small number of exclusively Asian participants.

  1. Have the authors measured and evaluated the MPOD and PROS between early-AMD fellow eyes and control eyes without retinal diseases?

Yes. We have already described this point in the original manuscript, and added a sentence for accuracy, as follows;

Line 96-

This study was conducted in the Medical Retina Division, Department of Ophthalmology, Keio University School of Medicine from March 2016 to April 2018 and included 30 unilateral AMD patients and 30 control patients who did not have retinal diseases or glaucoma. AMD patients with other retinal diseases or glaucoma were excluded.

Reviewer 2 Report

In this study the authors have examined the macular photopigment densitometry and photoreceptor outer-segment length of fellow eyes of 30 subjects with age related macular degeneration (AMD) (that are known to have increased risk of development of AMD), and have compared the results with age- and sex- matched control eyes (n = 30). It is suggested that a combination of macular photopigment densitometry and photoreceptor outer-segment length measurements can be a useful biomarker for screening eyes that are at risk of AMD. These results are novel and provide significant insights into potential mechanisms underlying processes leading to AMD. However, several methodological issues should be taken into consideration to have a better understanding of the accuracy/significance of the findings. In particular, the refractive state and time of day of measurements both can potentially impact the measurement of the photoreceptor outer-segment length (and also potentially the macular photopigment densitometry) and therefore confound the differences found between the AMD fellow eyes and the control group. Also, it is not clear whether the status and type of cataract have been controlled in the study, which may impact the measurements of threshold sensitivity for macular photopigment densitometry. Several other issues should also be discussed in the paper that have been raised in my comments as follows.  

  • Page 2 Line 53: “…eye involvement was 32% 2 years after…” suggest replacing “2” with “two”
  • Page 2 Line 62: “…energy-level short-wavelength light…” please specify the wavelength range here.
  • Page 2 Line 83: “…and/or pigment abnormality.” Suggest adding briefly here how the absence of pigment abnormality was confirmed in the study.
  • Page 3 Subjects: Please include information about the refractive error status of the subjects and whether the macular densitometry test was performed with or without optical correction?
  • Page 3 Subjects: Suggest including a statement on how the AMD (in the case groups) or absence of AMD (in the control group and in the fellow eye of AMD group) was diagnosed.
  • Page 3 Line 96: Did the authors control for the cataract status of the subjects, particularly given that the study focuses on elderly population with increased likelihood of cataract and the fact that cataract deteriorates the visual acuity/contrast sensitivity (e.g. https://www.ncbi.nlm.nih.gov/pmc/articles/PMC3306069/) and threshold sensitivity to short wavelength light (e.g. https://iovs.arvojournals.org/article.aspx?articleid=2161140)
  • Page 3 Line 96: It seems that whether the participants were implanted with intraocular lenses could impact the threshold sensitivity values, given the differences in the spectral absorption of IOLs. How was this controlled in the study?
  • Page 3 Line 97: Suggest including a brief statement about the direction of change expected to observe in MPOD with reduced magnitude of macular pigments (expected for AMD fellow eyes)
  • Page 3 Line 100: “Briefly, the difference in the intensity of blue…” Suggest replacing “intensity” with a word which implies the subjective nature of the measurement such as “threshold sensitivity”.
  • What was the wavelength of the blue and green flicker in MPOD
  • Suggest adding a explanation of the MPOD procedure used in the study - how many times was the test repeated? Was the non-testing eye occluded during the measurement? Was there any adaptation period observed prior to the measurements? 
  • Page 3 Line 106: Suggest including an OCT B-scan illustrating how PROS length was measured. It would also be good to include a B-scan from the AMD fellow eye and the control eye showing the difference in the PROS length between the two groups.  
  • Page 3 Line 107: Please clarify whether the PROS length was measured at the fovea, and (if yes) how the foveal position was defined.
  • Page 3 Line 107: Was there any consideration for the time of day of PROS length measurement, given that the thickness of the retinal outer segment changes diurnally by ~6 microns (e.g. https://iovs.arvojournals.org/article.aspx?articleid=2711344, and https://journals.lww.com/optvissci/FullText/2012/05000/Diurnal_Variation_of_Retinal_Thickness_with.12.aspx) and also the fact that PROS length differed between the two groups by a small magnitude of about 4 microns. 
  • Page 3: Data presented in table 1 is not well organised and difficult to follow. For BCVA only the range is included whereas for Age and CRT the range and mean (SD) are both included. For sex, it is not clear whether numbers are absolute or percentage frequency.
  • Given the role of the myoid and ellipsoid zone layer in photosensitivity, could there be any associations between the myoid and ellipsoid zone thickness and the MPOD in the AMD fellow eye and/or the control eyes?
  • Please check the references for consistent formatting

Author Response

Comments and Suggestions for Authors

In this study the authors have examined the macular photopigment densitometry and photoreceptor outer-segment length of fellow eyes of 30 subjects with age related macular degeneration (AMD) (that are known to have increased risk of development of AMD), and have compared the results with age- and sex- matched control eyes (n = 30). It is suggested that a combination of macular photopigment densitometry and photoreceptor outer-segment length measurements can be a useful biomarker for screening eyes that are at risk of AMD. These results are novel and provide significant insights into potential mechanisms underlying processes leading to AMD. However, several methodological issues should be taken into consideration to have a better understanding of the accuracy/significance of the findings. In particular, the refractive state and time of day of measurements both can potentially impact the measurement of the photoreceptor outer-segment length (and also potentially the macular photopigment densitometry) and therefore confound the differences found between the AMD fellow eyes and the control group. Also, it is not clear whether the status and type of cataract have been controlled in the study, which may impact the measurements of threshold sensitivity for macular photopigment densitometry. Several other issues should also be discussed in the paper that have been raised in my comments as follows.  

We appreciate the reviewer for understanding our work and providing constructive advice to improve our manuscript.

  • Page 2 Line 53: “…eye involvement was 32% 2 years after…” suggest replacing “2” with “two”

We revised accordingly.

  • Page 2 Line 62: “…energy-level short-wavelength light…” please specify the wavelength range here.

We inserted a phrase as follows:

Line 62-

…by absorbing high-energy short-wavelength light [14] (450–500 nm, with peak absorption at 460 nm) [15],…

  • Page 2 Line 83: “…and/or pigment abnormality.” Suggest adding briefly here how the absence of pigment abnormality was confirmed in the study.

We included this point in the revised manuscript as follows:

Line 82-

The AMD-fellow eyes in the current study did not include early AMD eyes characterized by drusen and/or pigment abnormality confirmed by medical records, fundus photographs, and OCT images.

  • Page 3 Subjects: Please include information about the refractive error status of the subjects and whether the macular densitometry test was performed with or without optical correction?

Thank you for pointing this out. Yes, densitometry test was performed with optical correction. We added the information as follows:

Line 111

MPOD measurement was performed with best correction,…

Regarding refractive error, we included the data in the revised Table 1, and added description in the text as follows:

Line 133-

There were no significant differences in age, sex, BCVA, refractive errors, or central retinal thickness between the two groups.

Table 1. Participants’ characteristics of control and AMD fellow eyes.

Control eyes

AMD-fellow eyes

P

n (eyes)

30

30

Age (years)

51-84 (69.2 ± 1.5)

50 -85 (68.2 ± 1.8)

0.678

Sex (men; eyes [%])

16 (53.3)

22 (73.3)

0.108

BCVA (LogMAR)

-0.08 - 0.05 (-0.06 ± 0.01)

-0.09 - 0.10 (-0.07 ± 0.01)

0.534

Refractive error (diopters)

5.88 - +3.50 (-0.88 ± 0.49)

-4.63 - +3.50 (0.21 ± 0.33)

0.071

CRT (μm)

194 -279 (226 ± 4)

120 - 270 (215 ± 5)

0.078

Data are presented as range (mean ± standard error); Two-tailed t-test. BCVA, best-corrected visual acuity; CRT, central retinal thickness.

  • Page 3 Subjects: Suggest including a statement on how the AMD (in the case groups) or absence of AMD (in the control group and in the fellow eye of AMD group) was diagnosed.

According to the reviewer’s advice, we added a sentence as follows:

Line 99-

Presence or absence of AMD and other diseases were confirmed by medical records including ophthalmologic examinations and previous angiographies (for the AMD patients), fundus photographs, and OCT images acquired at the time of this study.

  • Page 3 Line 96: Did the authors control for the cataract status of the subjects, particularly given that the study focuses on elderly population with increased likelihood of cataract and the fact that cataract deteriorates the visual acuity/contrast sensitivity (e.g. https://www.ncbi.nlm.nih.gov/pmc/articles/PMC3306069/) and threshold sensitivity to short wavelength light (e.g. https://iovs.arvojournals.org/article.aspx?articleid=2161140)

Yes, we excluded the patients who had severe cataract from the study. All the patients included in the current study had only mild nuclear and/or cortical cataract, and overall BCVA ranged -0.09 to 0.10 and mostly good. We included the sentence in the text as follows;

Line 102

Patients who had severe cataract were excluded from the study.

Line 135-

All the examined patients had mild nuclear and/or cortical cataracts, and no patient had highly myopic (<-6 diopters) or pseudophakic eyes.

  • Page 3 Line 96: It seems that whether the participants were implanted with intraocular lenses could impact the threshold sensitivity values, given the differences in the spectral absorption of IOLs. How was this controlled in the study?

Thank you for your comment. There were no patients who were implanted with intraocular lenses in the current study. This was included in the revised text as follows:

Line 135-

All the examined patients had mild nuclear and/or cortical cataracts, and no patient had highly myopic (<-6 diopters) or pseudophakic eyes.

  • Page 3 Line 97: Suggest including a brief statement about the direction of change expected to observe in MPOD with reduced magnitude of macular pigments (expected for AMD fellow eyes).

According to your advice, we included a sentence as follows:

Line 110-

Thus, if macular pigment levels were reduced, MPOD would be lower.

  • Page 3 Line 100: “Briefly, the difference in the intensity of blue…” Suggest replacing “intensity” with a word which implies the subjective nature of the measurement such as “threshold sensitivity”.

We revised the part accordingly.

  • What was the wavelength of the blue and green flicker in MPOD

The wavelength of blue light was 465 nm, and that of green light was 530 nm. We included the information in the text.

  • Suggest adding a explanation of the MPOD procedure used in the study - how many times was the test repeated? Was the non-testing eye occluded during the measurement? Was there any adaptation period observed prior to the measurements?

One test by using a series of stimuli to find the threshold was performed. There was no particular adaptation period but there was a preliminary measurement using 5 times of stimuli, and reliability of the measurement was confirmed automatically by the built-in program. The non-testing eye was occluded.

Line 111-

MPOD measurement was performed with best correction, and by using a series of stimuli programmed by the manufacturer. Non-testing eyes were occluded.

  • Page 3 Line 106: Suggest including an OCT B-scan illustrating how PROS length was measured. It would also be good to include a B-scan from the AMD fellow eye and the control eye showing the difference in the PROS length between the two groups.  

According to the reviewer’s suggestion, we added revised Figure 4 to show the difference in the PROS length between the two groups with arrowheads to show how PROS length was measured. 

Revised text and Figure 4 were as follows;

Line 181-

Figure 4 shows the representative OCT images of control and AMD-fellow eyes with MPOD.

Figure 4. Representative optical coherence tomography (OCT) images of control and age-related macular degeneration (AMD)-fellow eyes. (a) Control eye; a 61-year-old man whose photoreceptor outer segment (PROS) length was 49 μm and macular pigment optical density (MPOD) was 0.91. (b) AMD fellow eye; A 54-year-old man whose PROS length was 37 μm and MPOD was 0.53. PROS length is shown by the distance between two arrowheads.

  • Page 3 Line 107: Please clarify whether the PROS length was measured at the fovea, and (if yes) how the foveal position was defined.

We revised the text as follows:

Line 117-

OCT images were used to evaluate PROS length, which was defined as the distance between the inner border of the retinal pigment epithelium and the inner border of the ellipsoid zone at the fovea determined by the retinal depression in the horizontal images.

  • Page 3 Line 107: Was there any consideration for the time of day of PROS length measurement, given that the thickness of the retinal outer segment changes diurnally by ~6 microns (e.g. https://iovs.arvojournals.org/article.aspx?articleid=2711344, and https://journals.lww.com/optvissci/FullText/2012/05000/Diurnal_Variation_of_Retinal_Thickness_with.12.aspx) and also the fact that PROS length differed between the two groups by a small magnitude of about 4 microns. 

The measurement of PROS length was all performed in the OCT images taken during the Medical Retina Division Clinic from 13:00 to 17:00, and the data may have less influence of the measurement time. We included the time of measurement in the revised text as follows:

Line 116-

OCT was performed using a Heidelberg Spectralis OCT system (Heidelberg Engineering GmbH, Dossenheim, Germany) in the afternoon from 13:00 to 17:00.

  • Page 3: Data presented in table 1 is not well organised and difficult to follow. For BCVA only the range is included whereas for Age and CRT the range and mean (SD) are both included. For sex, it is not clear whether numbers are absolute or percentage frequency.

Thank you for your comment. We added some information in the footnote and in the table as shown above. The mean (SD) of BCVA had already been included in the original table, and we revised to show the range and mean (SD) in one line.

  • Given the role of the myoid and ellipsoid zone layer in photosensitivity, could there be any associations between the myoid and ellipsoid zone thickness and the MPOD in the AMD fellow eye and/or the control eyes?

According to your suggestion, we analyzed the associations between the myoid and ellipsoid zone thickness and the MPOD. While mean myoid and ellipsoid zone thickness in AMD-fellow eyes was significantly shorter than that in control eyes (AMD-fellow eyes, 32.6 ± 0.5 μm; Control 36.2 ± 0.4 μm, P < 0.001), there was no correlation between myoid and ellipsoid zone thickness and MPOD both in AMD-fellow eyes and control eyes (AMD-fellow eyes, R = 0.263, P = 0.159; Control R = 0.015, P = 0.939).

  • Please check the references for consistent formatting

Thank you for your comment. We checked the format.

Round 2

Reviewer 2 Report

I am still not convinced with how the foveal location was defined in the study. The authors state that the fovea has been determined as "the retinal depression in the horizontal images", which does not seem accurate. The retinal depression observed in a B-scan may simply represent the para-foveal region (and not the fovea), therefore any measurement of the PROS would be expected to be significantly lower than the foveal PROS. This is an important issue which may confound the PROS measurement, particularly in AMD fellow eyes in which the typical elongation of PROS in the foveal region (seen in normal control eyes) may not be easily detected (leading to an erroneous comparison of "naturally reduced" parafoveal PROS in AMD fellow eyes with "easily identifiable elongated" PROS in normal eyes of the control group).

The reviewer believes that the horizontal scan exhibiting the "deepest" retinal depression should have been selected to measure PROS in the fovea, which would clearly require a dense raster volumetric scan in both study groups. It is not clear whether such raster scan was available to the authors. 

The authors need to provide a robust definition of the fovea, otherwise, the PROS measurements provided would not be reliable and pose an important limitation to the study.  

Author Response

Point-by-Point Response to the Reviewer’ Comment

Reviewer 2

Comments and Suggestions for Authors

I am still not convinced with how the foveal location was defined in the study. The authors state that the fovea has been determined as "the retinal depression in the horizontal images", which does not seem accurate. The retinal depression observed in a B-scan may simply represent the para-foveal region (and not the fovea), therefore any measurement of the PROS would be expected to be significantly lower than the foveal PROS. This is an important issue which may confound the PROS measurement, particularly in AMD fellow eyes in which the typical elongation of PROS in the foveal region (seen in normal control eyes) may not be easily detected (leading to an erroneous comparison of "naturally reduced" parafoveal PROS in AMD fellow eyes with "easily identifiable elongated" PROS in normal eyes of the control group).

The reviewer believes that the horizontal scan exhibiting the "deepest" retinal depression should have been selected to measure PROS in the fovea, which would clearly require a dense raster volumetric scan in both study groups. It is not clear whether such raster scan was available to the authors. 

The authors need to provide a robust definition of the fovea, otherwise, the PROS measurements provided would not be reliable and pose an important limitation to the study.  

Thank you for your comment. We determined the fovea by observing the deepest depression by horizontal images using a dense raster three-dimensional volumetric scan. The three-dimensional scan (below left figure) was obtained in the colored squared area (below right figure).

The text was revised as follows;

Line 117-

The macular area was scanned, and acquired a dense volume scan (20 × 20°, 6 x 6 mm); 97 B-scans, each spaced 60 μm apart; an automatic real-time mean of 2 scans. The PROS length was defined as the distance between the inner border of the retinal pigment epithelium and the inner border of the ellipsoid zone at the fovea determined by the deepest retinal depression in the horizontal images of the three-dimensional dense volume OCT scan.
